# An experimental-mathematical approach to predict tumor cell growth as a function of glucose availability in breast cancer cell lines

**Jianchen Yang[1], Jack Virostko[2,3,4], David A. Hormuth, II[4,5], Junyan Liu[1], Amy Brock[1,3,4], Jeanne Kowalski[3,4], Thomas E. Yankeelov[1,2,3,4,5,6]** *

**1** Department of Biomedical Engineering, The University of Texas at Austin, Austin, Texas, United States of America, **2** Department of Diagnostic Medicine, The University of Texas at Austin, Austin, Texas, United States of America, **3** Department of Oncology, The University of Texas at Austin, Austin, Texas, United States of America, **4** Livestrong Cancer Institutes, The University of Texas at Austin, Austin, Texas, United States of America, **5** Oden Institute for Computational Engineering and Sciences, The University of Texas at Austin, Austin, Texas, United States of America, **6** Department of Imaging Physics, The University of Texas MD Anderson Cancer Center, Houston, Texas, United States of America

* thomas.yankeelov@utexas.edu

**Data Availability Statement:** 1) The data is available on Figshare https://doi.org/10.6084/m9.figshare.13017206.v1 2) The code for the BT-474

## Abstract

We present the development and validation of a mathematical model that predicts how glucose dynamics influence metabolism and therefore tumor cell growth. Glucose, the starting material for glycolysis, has a fundamental influence on tumor cell growth. We employed time-resolved microscopy to track the temporal change of the number of live and dead tumor cells under different initial glucose concentrations and seeding densities. We then constructed a family of mathematical models (where cell death was accounted for differently in each member of the family) to describe overall tumor cell growth in response to the initial glucose and confluence conditions. The Akaikie Information Criteria was then employed to identify the most parsimonious model. The selected model was then trained on 75% of the data to calibrate the system and identify trends in model parameters as a function of initial glucose concentration and confluence. The calibrated parameters were applied to the remaining 25% of the data to predict the temporal dynamics given the known initial glucose concentration and confluence, and tested against the corresponding experimental measurements. With the selected model, we achieved an accuracy (defined as the fraction of measured data that fell within the 95% confidence intervals of the predicted growth curves) of 77.2 ± 6.3% and 87.2 ± 5.1% for live BT-474 and MDA-MB-231 cells, respectively.

## 1. Introduction

The major source of energy for many cancer cells comes from a high rate of glycolysis followed by lactate fermentation in the cytosol, even in the presence of sufficient oxygen—a phenomenon known as the Warburg effect [1, 2]. This contrasts with normal cells that exhibit a comparatively low rate of glycolysis followed by oxidative phosphorylation in the mitochondria. Additionally, high concentrations of oxygen can lead to a reduction of glycolytic activity,

data is located here: https://doi.org/10.6084/m9.
figshare.14544462.v1 3) The code for the MDA-
MB-231 data is located here: https://doi.org/10.
6084/m9.figshare.14544450.v2.

**Funding:** This research was funded by the Cancer
Prevention and Research Institute of Texas (www.
cprit.state.tx.us) through RR160005 to T.E.Y, and
RR160093 to Dr. Gail Eckhardt (which partially
supports J.K.), and the National Institutes of Health
(www.nih.gov) through NCI U01CA174706,
U01CA142565, R01CA186193, and U01CA253540
to T.E.Y. T.E.Y is a CPRIT Scholar in Cancer
Research. The funders had no role in study design,
data collection and analysis, decision to publish, or
preparation of the manuscript.

**Competing interests:** The authors have declared
that no competing interests exist.

known as the Pasteur effect [3]. Also, an observation by Sonveaux [4] supports the claim that well-oxygenated tumor cells utilize lactate, leaving glucose available for hypoxic cells. This phenomenon has stimulated numerous efforts to investigate the underlying mechanisms [2, 5, 6] of altered metabolism and has identified potential targets including glucose transporters [7], lactate transporters [8], and enzymes like hexokinase and pyruvate kinase in the pathway of glycolysis [9, 10] for the development of new therapeutics. Efforts have been made to rigorously model the development of tumor subpopulations, nutrient dynamics, energy demands, and tumor-environment interactions with ordinary differential equations (ODEs) [11], partial differential equations (PDEs) [12–16], agent-based models [17], and game theoretical models [17–20]. For example, in the model developed by Mendoza-Juez *et al.* [11], tumor cells were divided into two subpopulations, the oxidative cells that undergo aerobic oxidation of glucose and glycolytic cells that undergo glycolysis and produce lactate, corresponding to an oxidative phenotype and a Warburg phenotype. Proliferation and conversion between the two subpopulations was described by a set of ordinary differential equations. This study also considered the nutrient concentrations of glucose and lactate as a result of consumption and production by tumor cells, which in return, can cause conversion between phenotypes. Mendoza-Juez *et al.* [11] further provided preliminary validation of their model by comparing it to metabolic data available from several previously published studies [4, 21, 22]. However, as no direct calibration of this model to experimental data was performed, it was not possible to capture specific parameter values that could be used to characterize cell lines [12, 17], or make predictions of tumor cell dynamics as a function of glucose availability or utilization. Additionally, the reliance on a large number of unmeasured parameters makes further applications challenging. Therefore, in this work, we aim to simplify this model with a smaller set of parameters that can be estimated or calibrated from experimental data and recast the associated models we developed with these estimates to predict tumor growth given initial conditions.

We designed a set of experiments employing time-resolved microscopy to track the temporal change of the number of live and dead tumor cells *in vitro* given a set of initial confluences (i.e., seeding density) and glucose concentrations. To quantitatively characterize those observations, we developed a family of mathematical models to describe the proliferation and death of tumor cells as a function of glucose availability and consumption. These models, which are based on those of Mendoza-Juez *et al.* [11], take the form of systems of nonlinear, ordinary differential equations to describe the collective temporal behavior of tumor cells. We aim to identify the most parsimonious model from that family to optimally characterize tumor cell growth as a function of glucose dynamics. After the optimal model is selected, we quantify the proliferation rate, death rate due to glucose depletion, death rate due to the bystander effect, and the consumption rate of glucose in a training set. We then use this calibrated model to predict tumor cell growth given prescribed initial conditions in a validation set.

## 2. Materials and methods

Throughout the following text, the reader is encouraged to refer to Fig 1 which provides an overview of the experimental and computational modeling components of the study.

### 2.1 Cell culture

We applied our experimental-mathematical approach in two breast cancer subtypes to quantitatively characterize cell types known to have distinct phenotypes, molecular profiles, and metabolic activities. Triple negative breast cancer [23] (TNBC) is defined by the absence of the expression of the estrogen, progesterone, and HER2 (human epidermal growth factor receptor 2) receptors, while in HER2+ breast cancer [24], HER2 is overexpressed.

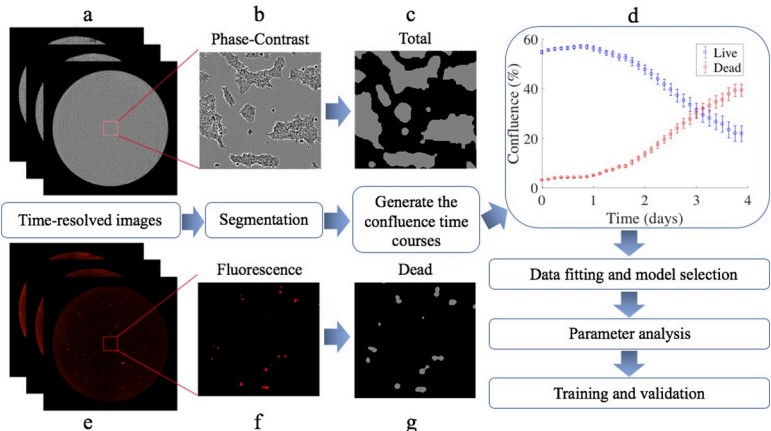

**Fig 1. A flow chart indicating the data acquisition and analysis steps.** Based on the phase-contrast (panels a and b) and fluorescent (panels e and f) images acquired from the time-resolved microscopy studies, we perform cell segmentation of total and dead cells (panels c and g, respectively) and generate time courses of confluence for both live and dead cells (panel d). The data are then used for selecting the most parsimonious mathematical model which estimates model parameters. Finally, the data are divided into subsets for training and validation of the predictive accuracy of the model.

BT-474 (a model of HER2+ breast cancer) and MDA-MB-231 (a model of triple negative breast cancer) cell lines were obtained from the American Type Culture Collection (ATCC, Manassas, VA) and maintained in culture according to ATCC recommendation. Ninety-six well-plates were seeded with either BT-474 or MDA-MB-231 cells at initial confluences ranging from 10% to 80% in Dulbecco's modified eagle medium (DMEM without glucose, sodium pyruvate, HEPES, L-glutamine and phenol red, Thermo Fisher Scientific, Waltham, MA) one day before imaging experiments began. On day zero, media were changed to DMEM with different glucose concentrations (0 mM, 0.1 mM, 0.2 mM, 0.5 mM, 0.8 mM, 1 mM, 2 mM, 5 mM, 8 mM and 10 mM). Each initial condition had four replicates. Cells were cultured in 5% $CO_2$ and air at 37˚C for 4 days.

## 2.2 Image acquisition

Cells were incubated in the IncuCyte S3 live cell imaging system (Essen BioScience, Ann Arbor, MI). Images were acquired with a 4× objective and stitched together to obtain whole well images (2400 × 2400 pixels) for each well of the 96-well plates *via* the device's whole-well imaging function. IncuCyte Cytotox Red Reagents (Essen BioScience, Ann Arbor, MI), a cyanine nucleic acid dye, was added to the medium on day 0 before the first scan to enable quantification of cell death. Once cells become unhealthy, the plasma membrane begins to lose integrity allowing entry of the IncuCyte Cytotox Reagent and yielding a 100-1000-fold increase in fluorescence upon binding to DNA. Phase-contrast and red fluorescent (excitation wavelength: 585 nm and emission wavelength: 635 nm) images were acquired every 3 hours for 4 days.

## 2.3 Image segmentation to quantify confluence over time

All cell segmentation was performed in Matlab (The Mathworks, Inc., Natick, MA). The segmentation approaches were developed based on the particular morphological features of the two cells lines. In particular, the BT-474 cells are mass cells with robust cell-cell adhesion that form cell clusters, while the MDA-MB-231 cells are elongated cells [25].

To segment the BT-474 cells within the phase-contrast images at each time point, a predetermined mask corresponding to the size of 96-well-plate from IncuCyte Software (Essen BioScience, Ann Arbor, MI) was first applied to the images so the region of interest (ROI) only included the area within each well and not the surrounding area of the plate in each square image. The masked image was then converted from the RGB (red, green, blue) format to grayscale and the Matlab function 'colfilt' was used to calculate the standard deviation of signal intensities within each 3-by-3 sliding block of the image to detect the edge of cell clusters. Next, a Gaussian filter was used to smooth the image returned from 'colfilt' to reduce the variance of signal intensities within each cell cluster. The resulting image was then normalized (by dividing the signal intensity of each pixel by the highest signal intensity from each image) between 0 and 1. After normalization, the morphological operator 'imerode' was used to make the clusters shrink in size and enlarge the holes to avoid losing open space within clusters. Next the returned image was converted to a binary image by the Matlab function 'im2bw'. The morphological operator 'imclose' was used to fill holes in the interior of cell clusters. The morphological operator 'imopen' was used to smooth object contours, break thin connections and remove thin protrusions. Finally, 'bwareaopen' was used to remove small objects like cell debris or noise. Please see S1 Fig for details and example images from each step.

While BT-474 cells form clusters that have clear boundaries, MDA-MB-231 cells are elongated and do not form clusters. This results in a much higher edge-area ratio in MDA-MB-231 images compared to BT-474. Thus, the segmentation scheme just described was adjusted to handle these differences in cell morphology. In particular, once the ROI was identified, 'histcount' was used to count the number of pixels for each signal intensity (256 possible signal intensity values in grayscale image) within the ROI. The pixels with signal intensities in the top 10% were assigned a 0, while the remaining pixels were assigned a 1 to binarize the image. All other steps were the same as the BT-474 segmentation. Please see S2 Fig for example images.

The fluorescent images were used to quantify the Cytotox Red signal (which marks the dead cells) for both cell lines. Since MDA-MB-231 cells change from an elongated to a circular morphology when they die, the differences in morphology of the two cell lines observed in phase-contrast images of the living cells vanishes. Thus, we applied the same approach segmenting the phase-contrast images of BT-474 cells to the florescent images of both cell lines.

The resulting segmented and binarized phase-contrast and fluorescent images were then analyzed to determine confluence at each time point. Confluence was defined as the percentage of the well covered by cells and was calculated by counting the number of pixels in the segmented images and dividing by the area of the field of view. Thus, our time-resolved microscopy data provided time courses of both living and dead cell number.

Tumor cell growth time courses were obtained from 4 experiments for each set of initial conditions, and each point in each time course consisted of a mean ± 95% confidence interval (a one-sample Kolmogorov-Smirnov test confirmed normality). One-way ANOVA was used to compare the average number of live cells for each experiment at the end of day 4 between the groups with different initial conditions.

## 2.4 Mathematical models

We developed a family of mathematical models to quantitatively and temporally describe the change in tumor cell number as function of glucose levels. To do so, we started with the model developed by Mendoza-Juez *et al.* [11] which describes the tumor as consisting of two subpopulations undergoing either aerobic oxidation of glucose or glycolysis, corresponding to Warburg and oxidative phenotypes, respectively. In our work, we first simplified the model to account for only one metabolic phenotype, and then extended it to account for the

Table 1. The definitions, units, and source for the model parameters.

| Parameter | Definitions | Units | Source |
|-----------|-------------|-------|--------|
| $k_p$ | Proliferation rate | day$^{-1}$ | Calibrated |
| $k_d$ | Death rate due to starvation | day$^{-1}$ | Calibrated |
| $k_{bys}$ | Death rate due to bystander effect | day$^{-1}$ | Calibrated |
| $\theta$ | Carrying capacity | cells | Assigned from literature [37] |
| $v$ | General glucose consumption | mM·cell$^{-1}$·day$^{-1}$ | Calibrated |
| $G^*$ | Michaelis-Menten constant | mM | Assigned from literature [11] |
| $G_{\min}$ | Minimum glucose level for uptake | mM | Assigned from literature [11] |

accumulation of dead tumor cells due to glucose depletion and the bystander effect [26, 27]. Accordingly, we modeled the change of glucose concentration as a result of consumption by all live tumor cells. Our complete model is described by a coupled set of ordinary differential equations shown below (the reader is encouraged to refer to Table 1 through the following discussion):

$$\frac{dN(t)}{dt} = k_p N(t)\left(1 - \frac{N(t)}{\theta}\right)S_p(G(t)) - k_d N(t)S_d(G(t)) - k_{bys}N(t)\left(\frac{D(t)}{D(t) + N(t)}\right) \quad [1]$$

$$\frac{dD(t)}{dt} = k_d N(t)S_d(G(t)) + k_{bys}N(t)\left(\frac{D(t)}{D(t) + N(t)}\right) \quad [2]$$

$$\frac{dG(t)}{dt} = -vN(t)\left(\frac{G(t)}{G(t) + G^*}\right), \quad [3]$$

where $N(t)$, $D(t)$, and $G(t)$ describe the live tumor cell number, dead tumor cell number, and glucose concentration, respectively, at time $t$. The first term on the right-hand side of Eq [1] describes logistic growth of tumor cells where $k_p$ is the proliferation rate, and $\theta$ is the carrying capacity. Here we define the carrying capacity as the limitation on the number of tumor cells that can physically fit within the environment. The logistic term is also modified by the state function, $S_p(G(t))$, that scales the proliferation rate as a function of glucose concentration. The second term on the right-hand side of Eq [1] describes the death of tumor cells due to glucose depletion at the rate $k_d$. This term is also modified by the state function, $S_d(G(t))$, that scales the rate of cell death as a function of glucose concentration. We assume that the dead tumor cells are accumulating and releasing factors [26, 27] which may be sensed by the remaining live cells and, potentially, induce cell death. This is referred to as the bystander effect [26, 27] and it is captured by the third term on the right-hand side of Eq [1] which induces cell death at the rate $k_{bys}$. The bystander effect may include the dead cells competing for space with live cells, cytotoxicity from dead cells, and increased acidity [28–30]. We assume the bystander effect is proportional to the fraction of dead cells. Eq [2] models the rate of change in number of dead cells, with the first term on the right-hand side describing death due to glucose depletion, and the second term accounting for the death due to the bystander effect. Eq [3] describes the change of glucose concentration due to the consumption by tumor cells at the rate $v$ and a Michaelis-Mentens constant, $G^*$. The state functions for tumor cell proliferation and tumor cell death are given as:

$$S_d(G(t)) = \left(1 - \frac{G(t)}{G(t) + G_{\min}}\right)\tanh(t) \quad [4]$$

$$S_p(G(t)) = 1 - \left(1 - \frac{G(t)}{G(t) + G_{min}}\right)\tanh(t), \tag{5}$$

where $G_{min}$ is the minimum glucose level required for proliferation. The parenthetical term on the right-hand side of Eq [4] describes the dependence of cell fate (proliferation or death) on glucose availability. In our approach, the proliferation (growth) rates and death rates should be considered as the maximum rates possible, while the real-time proliferation or death rates due to glucose depletion are modified by the state functions (Eqs [4] and [5]). According to Eq [5], as $G(t) \to \infty$, $S_p(G(t)) \to 1$ and the growth rate is maximized. Conversely, as $G(t) \to 0$, $S_p(G(t)) \to 0$ and the growth rate is minimized. That is, more glucose contributes to faster proliferation of cells. Similarly, according to Eq [4], the death rate due to glucose depletion is maximized as $G(t) \to 0$ and minimized as $G(t) \to \infty$. In summary, our model accounts for changes in both the growth and death rates due to glucose depletion as determined by the real-time glucose concentrations. As tumor cells may keep proliferating for some time even in a glucose free medium (please see S3 Fig), we introduced a hyperbolic tangent function of time. We hypothesize the tumor cell population is composed of two sub-populations, one that has passed the restriction point [31–34], is committed to divide, and thus does not need to be checked by the state function; and a second subpopulation that has not passed the restriction point, and thus has to be checked by the state function. The hyperbolic tangent function, tanh(t), increases from 0 to 1 as time increases from 0 to infinity; thus, tanh(t) on the right-hand side of Eq [4] introduces a delay due to the duration of mitosis [35, 36] of the cells that have passed the check point. All the cells were cultured in medium with the same glucose concentration (the "native" cell culture medium) after seeding (plating). Thus, all cell metabolism was initially driven by the same (native) glucose condition. After changing the media at the beginning of the experiment, the cells gradually switched to the second "state" where their metabolism were driven by the glucose concentrations in the media which were different than their native state. In our model, the tanh(t) term represents the time it takes for the cells to "switch" from the native state to the second "state". The number of cells that have passed the checkpoint before the medium change is determined by the glucose concentration in the "native" cell culture medium. This delay is not affected by the glucose concentration supplied in the medium after medium change. At time 0, the effect of glucose concentration described by the parenthetical term is multiplied by tanh(0), and becomes 0. This means the effect of glucose concentration is not sensed by cells immediately. At a later time, as tanh(t) increases to 1, the effect of glucose concentration increases until fully sensed by the cells. Afterwards, any further mitosis is fully affected by glucose concentration through the state function. The effect of the "native" cell culture media on the growth of the tumor cells would only exist at the earliest stages of the experiment, thereby making the tanh(t) function appropriate. Note that we have $S_d(G(t)) + S_p(G(t)) = 1$, as we assume the tumor cells are either proliferating or dying.

Eqs [1]–[5] can then be used to generate a family of three models by making a small set of simplifying assumptions. If we remove cell death due to the bystander effect in Eqs [1] and [2], we create another coupled system. Similarly, if we remove cell death due to glucose depletion in Eqs [1] and [2], we construct a third coupled system. Specifically, Model 2 is the complete model described by Eqs [1]–[5], Model 1 neglects cell death due to the bystander effect, and Model 3 neglects cell death due to glucose deprivation, but retains cell death due to the bystander effect. These three sets of equations provide our three-member model family which we then subject to a model selection operation to identify the most parsimonious model.

## 2.5 Model calibrations

The model described in the previous section was calibrated to experimentally measured live and dead cell time courses (described in Section 2.3), with the initial glucose concentration and confluence serving as the initial conditions. Recall that the overall goal was to calibrate model parameters against a test data set, and then use the subsequent parameterized model to predict tumor cell numbers in a validation cohort. To achieve this goal we performed a series of three calibrations for each cell line: one in which the parameters were calibrated for each individual time course, another in which the parameters were calibrated globally (i.e., a single set of parameters for the entire cohort/test set), and in the third in which we combined the results from the first two approaches so that some parameters were calibrated globally and others calibrated individually as a function of initial conditions.

In the first calibration scenario, the measured live and dead tumor cell time courses were independently fit to the model (i.e., Eqs [1]–[5]) to produce separate estimates for each model parameter within each cell line. We conducted model fitting like this for all 120 pairs of living and dead confluence time courses to generate 120 sets of estimates for local parameters. This approach would provide the lowest error (the difference between model fitting and measured data), but come with a high possibility of overfitting. The resulting parameter values were then further analyzed to determine if their value was a function of initial glucose level and confluence. In the second calibration scenario, the measured live and dead tumor cell time courses were fit by assuming model parameters were independent of initial conditions; i.e., a single set of global model parameters were determined to simultaneously fit all time courses (for each cell line). This approach assumed that the parameter values were not affected by initial conditions and are specific to each cell line. This approach provided the highest error, but least likely to overfit. We then systematically investigated a range of combinations of local and global parameters to achieve a balance between the fitting error and the risk of overfitting. We investigated the distributions of estimates of the local parameters in different combinations. We noticed that the estimates of $k_p$, $k_d$, and $v$ tended to converge to similar values across replicates, suggesting they could be global parameters, while the estimates of $k_{bys}$ presented a wide distribution. Therefore, in the third calibration scenario, we assumed (based on the results of the first two calibration scenarios) that the proliferation rate, $k_p$, the consumption rate of glucose, $v$, and the death rate due to glucose depletion, $k_d$, were specific for each cell line, while the other parameter, $k_{bys}$ was a function of initial confluence and glucose levels. A Student's $t$-test was used to test for statistical differences, between the two cell lines, of each global model parameter (i.e., $k_p$, $v$, and $k_d$) estimated.

To perform each of the above calibrations, we employed a non-linear, least squares approach which seeks to minimize the residual sum of square (RSS) errors between the measured data and the model described in section 2.4. We defined the system of ODEs, initial conditions, and time steps in Matlab using the built-in ODE solver 'ode45' to estimate the model parameters. We used a least square optimization algorithm 'lsqcurvefit' to update the parameter estimates and minimize the RSS errors. To avoid local minima, we used Matlab's 'MultiStart' to run, in parallel, 10 optimization problems with different initial parameter guesses to identify the set of parameters that minimized the RSS error. The initial parameter guesses that led to the solution point with the lowest (best) RSS error were recorded and set to be the single-start initial points for a second round of 'lsqcurvefit' to calculate the residuals and Jacobian matrix, which cannot be acquired during the first-round fitting with multiple starting points. The residuals and Jacobian matrix were used to determine the confidence interval for each parameter by calling the function, 'nlparci'. Before the fitting procedure, the initial live and dead tumor cell numbers were assigned as the average of the first three timepoints to reduce error in the estimation of the initial conditions.

## 2.6 Model selection

As the three models (described in Section 2.4) with the different fitting strategies (described in Section 2.5) are phenomenological in nature (i.e., they are not derived from first principles), we do not know which one, *a priori*, provides the best description of the experimental data. To address this limitation, we performed model selection *via* the Akaike Information Criteria (AIC) [38]. The AIC seeks to select the most parsimonious model by balancing goodness of fit with the number of free parameters. Given our data set, we will employ the $AIC_c$ [39, 40] which includes a correction for small sample size and is given as follows:

$$AIC_c = n\ln(RSS) + 2p + \frac{2p(p+1)}{n-p-1}, \qquad [6]$$

where $n$ is the number of data samples and $p$ is the number of model parameters. The model with the lowest $AIC_c$ value is selected as the most parsimonious.

## 2.7 Determining the dependence of model parameters on initial conditions

We investigated the correlation between the model parameters and the initial conditions. If a relationship can be found between a given parameter and initial conditions, then that parameter can be assigned on an individual experimental basis *via* an experimentally developed "look-up" table. Such an approach helps to "personalize" each prediction as these model parameters are determined by the initial condition of the experiment under investigation. Furthermore, the initial conditions are (by definition) frequently known (or, at least, bounded) at the beginning of the experiment so it provides a practical way to constrain model predictions. The results of the third calibration scenario showed that $k_{bys}$ for the BT-474 line increased with higher initial confluence (see S4 Fig), but decreased with higher initial glucose level, while $k_{bys}$ for the MDA-MB-231 line was not affected by initial confluence (see S5 Fig), but decreased with higher initial glucose level. The dependence of local parameter (i.e., parameters calibrated for individual time courses) values on initial conditions were determined by Pearson's partial correlation coefficient. Given this relationship, we sought to determine if there was a simple functional relation between model parameters and initial conditions. We were able to find one such relation for $k_{bys}$ for the BT-474 cells:

$$k_{bys} = k_{bys,0}N_0\exp(-\alpha G_0), \qquad [7]$$

where $N_0$ is the initial confluence, $G_0$ is the initial glucose concentration, $k_{bys,0}$ is the maximum $k_{bys}$ rate, and $\alpha$ is a decay parameter. We then fit Eq [7] to the set of initial conditions and associated parameter estimates (with their confidence intervals) obtained from the training data set to estimate $k_{bys,0}$, $\alpha$, and their respective 95% confidence interval. Thus, Eq [7] determines a parameter surface where $k_{bys}$ can be estimated given the initial confluence and glucose. This death rate, combined with estimates of the other global parameters (i.e., $k_p$, $v$, and $k_d$), can then be substituted into the Eqs [1]–[5] to predict tumor cell number at future time points. Using an analogous procedure, a similar relation was determined for the MDA-MB-231 cells:

$$k_{bys} = k_{bys,0}\exp(-\alpha G_0) + \beta, \qquad [8]$$

where the parameters are as indicated for Eq [7], with $\beta$ being a base death rate which is present in this cell line even when sufficient glucose is present. Note that $N_0$ does not appear in Eq [8] as this death rate for MDA-MB-231 cells is not affected by the initial confluence. Thus, Eq [8] also determines a parameter curve where $k_{bys}$ can be estimated given the initial glucose level. Again, this death rate, combined with estimates of the other global parameters (i.e., $k_p$, $v$,

and $k_d$), can then be substituted into the Eqs [1]–[5] to predict tumor cell number at future time points.

## 2.8 Training and validation

The data measured from the time-resolved microscopy experiments were divided into training (75% of the data) and validation sets [41, 42] by random sampling. The 120 pairs (each pair in our experiments consisted of a confluence time course for live cells and a confluence time course for dead cells from the same well) of confluence time courses were numbered from 1 to 120. In each round of training, a random number generator (Matlab function 'randperm') was used to randomly select 90 integers from the integers 1 to 120 without repeats. The confluence time courses with the corresponding number were used as the training set (75% of the data), while the remaining 30 sets of confluence time courses were used as the validation set (25% of the data). The training subset was used to calibrate the global parameters $k_p$, $k_d$, and $\nu$. We calculated the absolute value of the error between the best fit curve and measured data across the whole training set to provide an estimate of the error in the measurement (i.e., uncertainty) of the initial number of live and dead tumor cells, as required for forming a prediction on the validation set. Then, given these global parameters, and the initial conditions (i.e., $G_0$ and $N_0$) from each time course in the validation set, $k_{bys}$ was calculated using Eq [7] for the BT-474 line or [8] for the MDA-MB-231 line. Next, $k_{bys}$ was combined with the global parameters and initial conditions to run the forward model *via* Eqs [1]–[5]. The resulting predicted tumor cell number time courses (with confidence intervals) for live and dead tumor cells were compared to the corresponding measured data and the errors were tabulated. We defined 'prediction accuracy' as the fraction of measured data that fell within the 95% confidence intervals of the predicted growth curves, while accuracy for the whole validation set was determined as the average 'prediction accuracy' over all measured time courses. This training and validation process was repeated 50 times, and the average error for predicted time courses and average overall accuracy was recorded. To evaluate the model's performance, we report the averages of the RSS, mean percent error over the time course percent error at the end of experiment, mean error over the time course, error at the end of experiment (explicit matrices are presented in S2 Table).

## 3. Results

### 3.1 Tumor cell growth with different initial conditions

Example time courses for the BT-474 cell line, with different initial confluences (i.e., seeding density) and four glucose concentrations are shown in Fig 2A–2C. For wells with low initial confluence (Fig 2A), intermediate initial confluence (Fig 2B), and high initial confluence (Fig 2C), the percent change (mean ± 95% confidence interval) on the number of live cells from day 0 to day 4 with initial glucose concentrations of 0.2 mM, 0.5 mM, 2 mM, and 5 mM are shown in Table 2. The average number of live cells for each experiment at the end of day 4 was significantly different among the groups with different initial conditions ($p < 10^{-5}$).

Example time courses for the MDA-MB-231 cell line, with different initial confluences and four glucose concentrations, are shown in Fig 2D–2F. For wells with low initial confluence (Fig 2D), intermediate initial confluence (Fig 2E), and high initial confluence (Fig 2F), the percent change on the number of live cells from day 0 to day 4 with initial glucose concentrations of 0.2 mM, 0.5 mM, 2 mM, and 5 mM are shown in Table 2. The average number of live cells for each experiment at the end of day 4 was significantly different among the groups with different initial conditions ($p < 10^{-5}$).

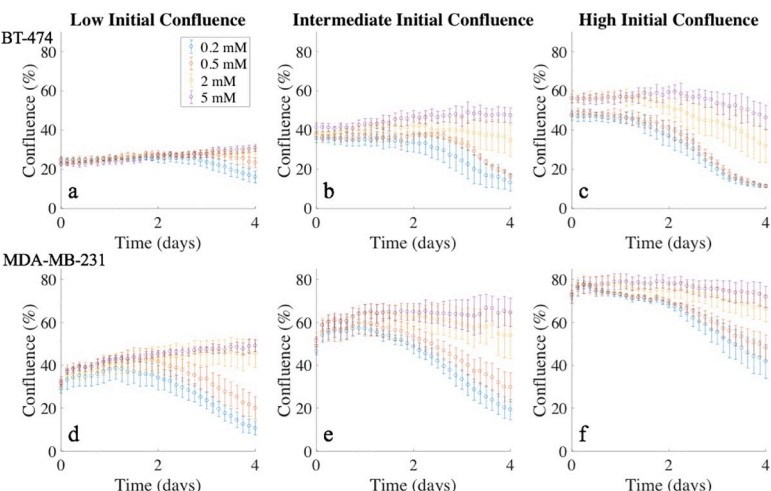

**Fig 2. Time courses of tumor cell confluence in media with varying initial glucose levels, grouped by initial confluence.** Panels a-c present confluence time courses for the BT-474 cell line, while panels d-f present confluence time courses for the MDA-MB-231 cell line. Different colors represent the four initial glucose concentrations, and the error bars were calculated from four replicates with similar initial conditions. In each panel, cells represented by each color were seeded at the same initial confluence, but yielded significant differences in confluence at the end of the experiment. These time courses provide quantitative and dynamic data on the effects of glucose availability and confluence on tumor cell growth.

## 3.2 Model calibration

The model characterized by Eqs [1]–[5] featuring three global parameters ($k_p$, $k_d$, and $v$), and one local parameter dependent on initial conditions ($k_{bys}$) was selected by the $AIC_c$ as the most parsimonious and employed for all subsequent analysis (details provided in S1 Table). The estimates for the three global parameters and their 95% confidence intervals for both BT-474 and MDA-MB-231 cells are shown in Table 3. The proliferation and glucose consumption rates of the BT-474 cells were significantly lower than the MDA-MB-231 cells ($p < 10^{-4}$), while the death rate due to glucose depletion of the BT-474 cells was higher than MDA-MB-231 cells ($p < 10^{-4}$).

As described in Section 2.5, the selected model (i.e., the model with globally calibrated $k_p$, $k_d$, and $v$ and locally calibrated $k_{bys}$) was fit to the measured experimental data. The mean percent error across all timepoints, mean percent error at the end of experiment mean error across all timepoints, and mean error at the end of experiment are reported in Table 4. The model was able to provide an accurate description of the time course data over a wide range of initial conditions with mean percent error and mean percent error at the end of experiments below 7% for live cells in both cell lines (Table 4). For the dead cells, the model performs more

**Table 2. Average percent change on the number of live cells from day 0 to day 4.**

| Cell Line | Initial Confluence (%) | | Initial Glucose Concentration (mM) | | | |
|---|---|---|---|---|---|---|
| | | | 0.2 | 0.5 | 2 | 5 |
| BT-474 | Low | 23.8 ± 0.5 | -34.3 ± 12.3 | -6.5 ± 10.5 | +31.4 ± 8.4 | +35.7 ± 1.8 |
| | Intermediate | 35.9 ± 1.8 | -63.7 ± 9.3 | -55.6 ± 3.1 | -10.4 ± 19.6 | +14.9 ± 8.3 |
| | High | 51.7 ± 1.4 | -76.0 ± 1.0 | -76.3 ± 1.5 | -43.9 ± 12.4 | -17.6 ± 7.6 |
| MDA-MB-231 | Low | 36.9 ± 1.0 | -68.3 ± 7.9 | -47.7 ± 12.7 | +30.2 ± 13.5 | +32.9 ± 7.9 |
| | Intermediate | 56.2 ± 1.4 | -63.7 ± 10.3 | -46.5 ± 10.6 | -1.8 ± 3.4 | +13.3 ± 2.1 |
| | High | 71.9 ± 1.0 | -41.5 ± 11.6 | -33.6 ± 9.2 | -10.0 ± 3.2 | -1.6 ± 2.9 |

**Table 3. Parameter estimates obtained from the global calibration procedure.**

| Parameters | Cell Line | | p-value |
|---|---|---|---|
| | **BT-474** | **MDA-MB-231** | |
| $k_p$ (day$^{-1}$) | 0.092 ± 0.002 | 0.14 ± 0.003 | $< 10^{-4}$ |
| $k_d$ (day$^{-1}$) | 0.13 ± 0.013 | 0.041 ± 0.006 | $< 10^{-4}$ |
| $v$ (×$10^{-5}$ mM·cell$^{-1}$·day$^{-1}$) | 2.68 ± 0.10 | 4.48 ± 0.15 | $< 10^{-4}$ |

modestly with mean percent error between 16% and 67% over all initial conditions. Importantly, the mean error, either across all timepoints or at the end of experiment, was < 2% for both live and dead cells in both cell lines. This suggests the higher percent error of dead cells is due to the small number of dead cells as compared to the number of live cells.

### 3.3 Relationship between bystander effect death rate ($k_{bys}$) and initial conditions

For the BT-474 cells, the death rate due to the bystander effect, $k_{bys}$, was found to increase with increasing initial confluence, with a partial correlation coefficient of 0.66 (p $< 10^{-4}$). For 8 of 10 initial glucose levels tested (0, 0.1, 0.2, 0.5, 0.8, 1, 2, and 5 mM), the bystander effect death rate was positively correlated with initial confluence, with correlation coefficients all above 0.74 (p < 0.01). Estimates of $k_{bys}$ were plotted against the initial confluence level (Fig 3A). The correlation coefficients between $k_{bys}$ and initial confluence level were 0.75, 0.94, 0.81, 0.81, and 0.84 for initial glucose level of 0.2 mM, 0.5 mM, 1 mM, 2 mM, and 5 mM, respectively. For the highest two initial glucose levels (8 and 10 mM), there was no significant correlation between the bystander effect death rate and the initial confluence (p > 0.1). The bystander effect was found to decrease with increasing initial glucose level, with a partial correlation coefficient of -0.74 (p $< 10^{-4}$). For low (23.8 ± 0.5%), intermediate (35.9 ± 1.8%), and high (51.7 ± 1.4%) initial confluences, there are significant correlations between $k_{bys}$ and the initial glucose level (Fig 3B), with correlation coefficient of -0.44 (p < 0.01), -0.80 (p $< 10^{-4}$), and -0.91 (p $< 10^{-4}$). Given these relationships, $k_{bys}$ was fit to each initial condition as described in Section 2.7 (see Eq [7]), yielding a $k_{bys,0}$ of 2.37 ± 0.13 × $10^{-5}$ mM·cell$^{-1}$·day$^{-1}$ and an $\alpha$ of 0.13 ± 0.029 mM$^{-1}$. With $k_{bys,0}$ and $\alpha$ identified, Eq [7] defines a parameter surface where we can obtain the value of $k_{bys}$ for any initial confluence and glucose level within the experimentally measured range (Fig 3D).

For the MDA-MB-231 cells, there was no significant correlation between the death rate due to the bystander effect and the initial confluence with a partial correlation coefficient of -0.03 (p = 0.73). The bystander effect was found to decrease with increasing initial glucose level

**Table 4. Evaluation of fitting quality with selected model for both cell lines.**

| | Cell Line | | | |
|---|---|---|---|---|
| | **BT-474** | | **MDA-MB-231** | |
| | **Live** | **Dead** | **Live** | **Dead** |
| **RSS** | **1.35** | **1.13** | **1.87** | **1.31** |
| Mean % Error | 0.09 ± 0.23 | 66.01 ± 4.13 | 0.59 ± 0.22 | 18.17 ± 1.66 |
| % Error EoE* | -0.78 ± 3.44 | 44.37 ± 30.63 | 6.22 ± 2.31 | 16.92 ± 18.43 |
| Mean Error | -0.01 ± 0.06 | 0.61 ± 0.05 | -0.03 ± 0.07 | 0.87 ± 0.05 |
| Error EoE* | 0.24 ± 0.57 | 0.56 ± 0.42 | 1.65 ± 0.42 | 0.42 ± 0.43 |

*EoE = End of Experiment

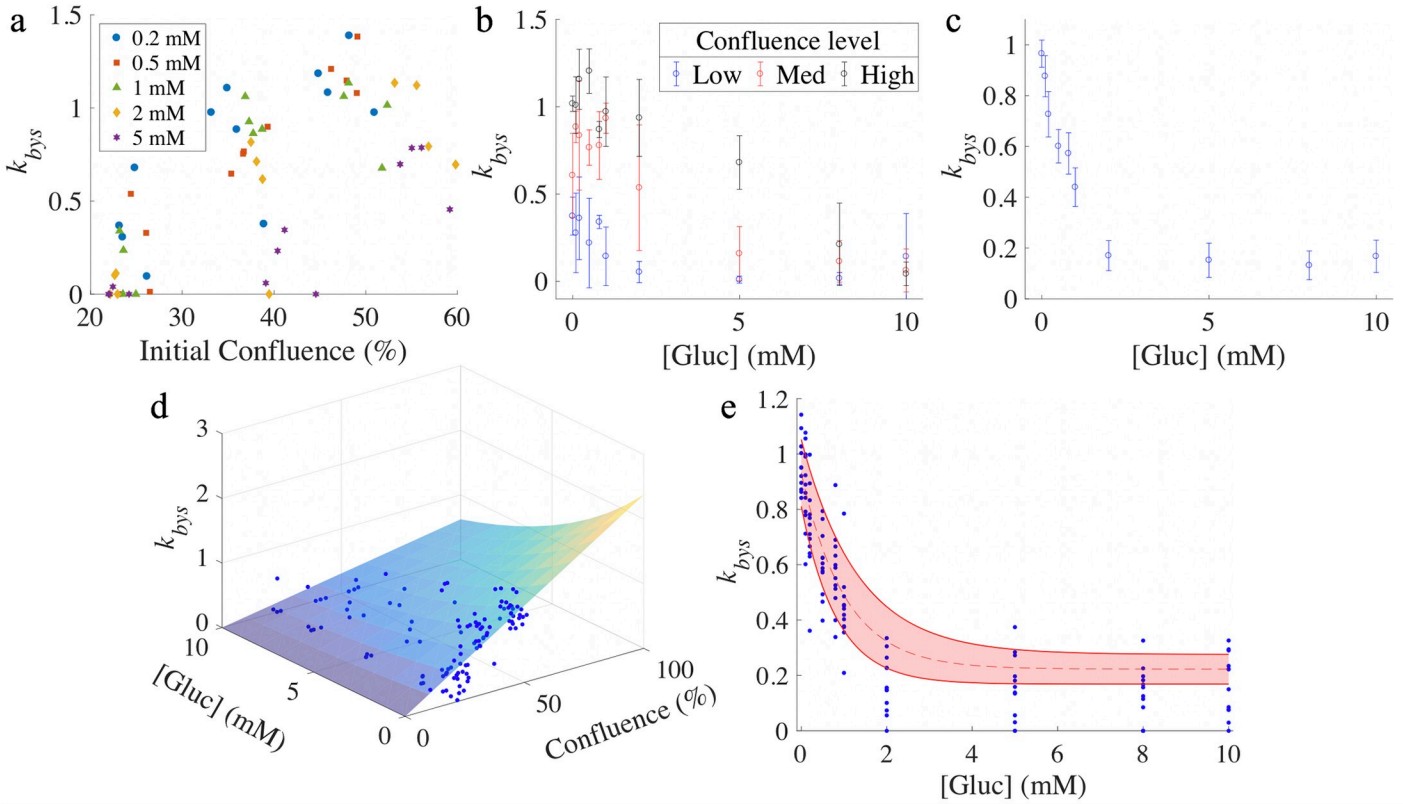

**Fig 3. Relationship between bystander effect death rate ($k_{bys}$) and initial conditions.** Panel a presents estimates of the death rate due to the bystander effect, $k_{bys}$, as a function of different initial confluence and glucose levels for BT-474 cells. For each glucose level, $k_{bys}$ increases with higher initial confluence, where the lowest initial glucose level increases $k_{bys}$ the most. Panel b indicates that $k_{bys}$ increases with higher initial confluence and decreases with higher initial glucose level for the BT-474 line. (Error bars were calculated from the four wells with similar initial conditions.) Panel c shows that $k_{bys}$ decreases with higher initial glucose level for the MDA-MB-231 cell line. (Error bars were calculated from the twelve wells with same initial glucose level.) Panel d shows the parameter surface for the BT-474 cell line, where $k_{bys}$ is displayed as function of initial confluence and glucose level, with blue dots representing calibrated estimates of $k_{bys}$. Panel e indicates how $k_{bys}$ decreases with initial glucose level for the MDA-MB-231 line, with shaded area between solid red curves showing the 95% confidence interval. The blue dots represent the calibrated estimates of $k_{bys}$. The fitted surface and curve in panels d and e, respectively, is used to assign $k_{bys}$ as a function of initial confluence and glucose concentration in the validation data set.

(Fig 3C), with a partial correlation coefficient of -0.72 ($p < 10^{-4}$). For the low (36.9 ± 1.0%), intermediate (56.2 ± 1.4%), and high (71.9 ± 1.0%) initial confluences, there are significant correlations between the death rate due to the bystander effect and the initial glucose level, with correlation coefficient of -0.76 ($p < 10^{-4}$), -0.76 ($p < 10^{-4}$), and -0.66 ($p < 10^{-4}$). Given these relationships, $k_{bys}$ was fit to each initial condition as described in Section 2.7 (see Eq [8]), yielding a $k_{bys,0}$ of 0.71 ± 0.067 × $10^{-5}$ mM·cell⁻¹·day⁻¹, an $\alpha$ of 0.98 ± 0.23 mM⁻¹ and a $\beta$ of 0.22 ± 0.053 mM·cell⁻¹·day⁻¹. With $k_{bys,0}$, $\alpha$, and $\beta$ identified, Eq [8] defines a parameter curve where we can obtain the value of $k_{bys}$ for any initial glucose level within the experimentally measured range (Fig 3E).

### 3.4 Evaluation of model performance through training and validation

In each round of training and validation, 75% of the whole dataset was randomly selected for a training set, with the remainder assigned to the validation set. The selected model (i.e., the model with globally calibrated $k_p$, $k_d$, and $\nu$ and locally calibrated $k_{bys}$) was calibrated to each time course in the training set to obtain estimates and confidence intervals for the model parameters.

**Table 5. Summary of model calibration across 50 training sets.**

| | Cell Line | | | |
| --- | --- | --- | --- | --- |
| | BT-474 | | MDA-MB231 | |
| | Live | Dead | Live | Dead |
| RSS | 1.09 ± 0.02 | 0.88 ± 0.01 | 2.22 ± 0.02 | 0.96 ± 0.02 |
| Mean % Error | 0.24 ± 0.04 | 72.42 ±1.23 | -0.26 ± 0.06 | 12.29 ± 0.29 |
| % Error EoE* | -0.90 ± 0.28 | 49.88 ± 2.41 | 6.01 ± 0.24 | 11.25 ±1.62 |
| Mean Error | 0.07 ± 0.01 | 0.69 ± 0.01 | -0.52 ± 0.03 | 0.19 ± 0.02 |
| Error EoE* | 0.25 ± 0.04 | 0.69 ±0.03 | 1.44 ±0.05 | -0.62 ± 0.05 |
| Uncertainty | 6.88 ± 0.09 | 30.83 ± 0.15 | 5.17 ± 0.05 | 16.78 ± 0.12 |

*EoE = End of Experiment

For the BT-474 cells, we reported the model performance during training (Table 5). The average mean percent error across all timepoints, and the average percent error at the end of experiment were < 1% for live cells. Although the average mean percent error across all time-points, and the average percent error at the end of experiment were > 45% for dead cells, the average mean error across all timepoints and average error at the end of experiment were < 1% for both live and dead cells. The average uncertainty across 50 training sets for live and dead cells were 6.88 ± 0.09% and 30.83 ± 0.15%, respectively.

For the BT-474 cells, the parameters $k_{bys,0}$ and $\alpha$ in Eq [7] were estimated as described in section 2.7 and a specific parameter surface of $k_{bys}$ was determined. The uncertainty calculated from fitting the data of the training set to the model was used to estimate the confidence interval of the initial confluence from the validation set. The initial conditions (i.e., initial glucose level and confluence) from the validation set were used with Eq [7] to identify the value of $k_{bys}$ to be used, in conjunction with the three global parameters ($k_p$, $v$, and $k_d$ and their respective confidence intervals) in Eqs [1]–[5] to run the forward model. This process was repeated 50 times to obtain an average RSS, average mean percent error, average percent error at the end of experiment, average mean error, average error at the end of experiment, and accuracy (Table 6). The accuracy was defined as the percent of data points falling within the 95% confidence interval of the predicted values. The average RSS was 1.45 ± 0.09 and 1.22 ± 0.09 for live and dead cells, respectively, while the accuracy was 77.2 ± 6.3% and 50.5 ± 7.5% for live and dead cells, respectively. The average mean percent error across all timepoints and the average percent error at the end of experiment were both < 2% for live cells. Although the average mean percent error across all timepoints and average percent error at the end of the

**Table 6. Evaluation of prediction across 50 rounds of training and validation.**

| | Cell Line | | | |
| --- | --- | --- | --- | --- |
| | BT-474 | | MDA-MB231 | |
| | Live | Dead | Live | Dead |
| RSS | 1.45 ± 0.09 | 1.22 ± 0.09 | 1.69 ± 0.10 | 1.35 ± 0.12 |
| Mean % Error | -1.96 ± 0.54 | 153.18 ± 9.07 | -0.59 ± 0.47 | 25.22 ± 1.27 |
| % Error EoE* | -5.78 ±1.49 | 168.20 ±16.31 | 7.04 ± 1.62 | 47.54 ± 9.95 |
| Mean Error | -0.78 ± 0.15 | 1.57 ± 0.13 | -1.15 ± 0.15 | 0.87 ± 0.16 |
| Error EoE* | -1.66 ± 0.33 | 2.67 ± 0.27 | -0.12 ± 0.38 | 1.09 ± 0.43 |
| Accuracy | 77.2 ± 6.3 | 50.5 ± 7.5 | 87.2 ± 5.1 | 66.7 ± 7.0 |

*EoE = End of Experiment

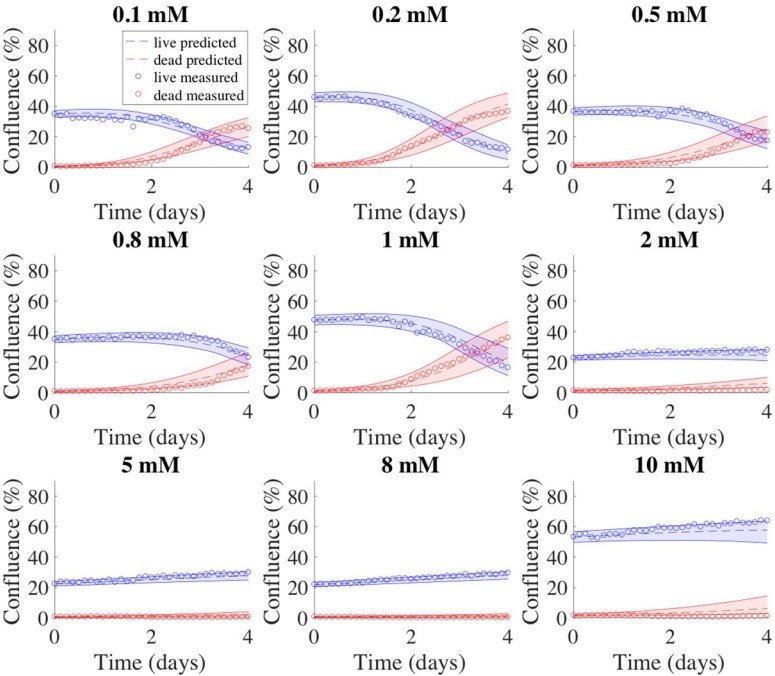

**Fig 4. Model predictions for BT-474 cells.** Example model predictions from one validation set of BT-474 cells are shown in dashed lines. In each panel, data measured from experiments are shown in circles, while the 95% confidence intervals for the predicted tumor cell growth and dead cell accumulation numbers are shown with shaded regions between the solid curves; with blue indicating live cells, and red indicating dead cells. The initial glucose level is shown above each plot. For this validation set, the model prediction accuracy was 72.2 ± 8.6% and 49.3 ± 10.0% for live and dead cells, respectively.

experiment for dead cells can be as high as > 150%, the average mean error across all time-points and average error at the end of experiment were < 3% for both live and dead cells. Fig 4 presents representative prediction results compared with measured data on BT-474 cells from the same round of training and validation (Fig 4).

For the MDA-MB-231 cells, we reported the model performance during training (Table 5). The average mean percent error across all timepoints, and the average percent error at the end of experiment were < 13% for both live and dead cells. The average mean error across all time-points and average error at the end of experiment were < 2% for both live and dead cells. The average uncertainty across 50 training sets for live cells and dead cells were 5.17 ± 0.05% and 16.78 ± 0.12% respectively.

For the MDA-MB-231 cells, the parameters $k_{bys,0}$, $\alpha$, and $\beta$ in Eq [8] were estimated as described in section 2.7 and a specific parameter curve for $k_{bys}$ was determined. The uncertainty calculated from fitting the data of the training set to the model was used to estimate the confidence interval of the initial confluence from the validation set. The initial condition (i.e., initial glucose level) from the validation set were used with Eq [8] to identify the value of $k_{bys}$ to be used, in conjunction with the three global parameters ($k_p$, $v$, and $k_d$ and their respective confidence intervals) in Eqs [1]–[5] to run the forward model. This process was repeated 50 times to obtain average RSS, average mean percent error, average percent error at the end of experiment, average mean error, average error at the end of experiment, and accuracy (Table 6). The accuracy was defined as the percent of data points falling within the 95% confidence interval of the predicted values. The average RSS was 1.69 ± 0.10 and 1.35 ± 0.12 for live and dead cells, respectively, while the accuracy was 87.2 ± 5.1% and 66.7 ± 7.0% for live and

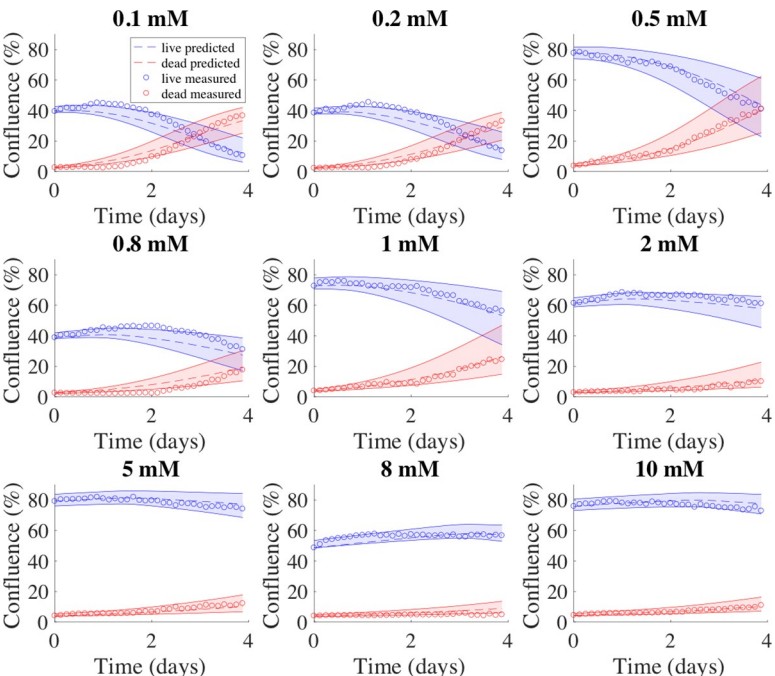

**Fig 5. Model predictions for MDA-MB-231 cells.** Example model predictions from one validation set of MDA-MB-231 cells are shown in dashed lines. In each panel, data measured from experiments are shown in circles, while the 95% confidence intervals for the predicted tumor cell growth and dead cell accumulation numbers are shown with shaded regions between the solid curves; with blue indicating live cells, and red indicating dead cells. The initial glucose level is shown above each plot. For this validation set, the model prediction accuracy was 86.9 ± 7.4% and 69.6 ± 9.4% for live and dead cells, respectively.

dead cells, respectively. The average mean percent error across all timepoints and the average of percent error at the end of experiment were both < 8% for live cells. Although the average percent error across all timepoints and average error at the end of experiment for dead cells were > 25%, the average mean error across all timepoints and average error at the end of experiment were < 2% for both live and dead cells. Fig 5 presents representative prediction results compared with measured data on MDA-MB-231 cells from the same round of training and validation (Fig 5).

## 4. Discussion

This study sought to develop an experimental-mathematical approach to quantify tumor cell proliferation as a function of glucose availability. This allowed us to quantify important cell phenotypes related to proliferation and cell death, and then use the model to predict the temporal change in tumor cell number. To accomplish this task, we proposed a family of three models in which each member of the family consisted of a system of coupled ordinary differential equations (ODEs) describing the rate of change of living and dead tumor cell number and glucose concentration. The complete model considered tumor cell proliferation, cell death due to glucose depletion, the bystander effect quantifying the effects of dead cells accumulated in the environment, and the consumption of glucose. To calibrate the model, we acquired time-resolved microscopy images to generate confluence time courses of both live and dead tumor cells over an array of initial glucose concentrations and confluences. We then fit the data to all the models and selected the most parsimonious model with the lowest $AIC_c$ value. The complete model (model 2; i.e., Eqs [1]–[5]) achieved the lowest fitting error for the

complete dataset, while the other two models yielded a significantly higher fitting error to the data. For model 1, which neglects the bystander effect, the death due to glucose depletion is not sufficient to capture all the cell death, especially for cells seeded with high initial glucose and high initial confluence. For model 3, which neglects the death due to glucose depletion, the initiation of cell death cannot be explained when there are initially no dead cells and, therefore, perform worse and overestimate the death rate for cells with low glucose level and few dead cells at the beginning. During the model selection, we determined that the proliferation rate, death rate due to glucose depletion, and consumption rate of glucose were three parameters that depended only on cell line and not initial conditions; thus, they could be fit as global parameters. Conversely, it was determined that the death rate due to the bystander effect was a local parameter that varied with the initial conditions. We therefore investigated the relationship between this parameter and the initial conditions for each cell line. The contributions from each physical term in the governing equations depend on initial conditions and change over time. In general, the contribution from logistic growth is dominant at the beginning and drops over time with the consumption of glucose. The contribution from death due to glucose depletion begins at time 0 and increases over time with the consumption of glucose. The contribution from death due to the bystander effect increases over time with the accumulation of dead cells. However, if there is sufficient glucose throughout the experiment, the contribution from logistic growth would remain dominant, with little contribution from death terms. Examples of relative contributions are provided in S3 (for BT-474 cells) and S4 Tables (for MDA-MB-231 cells). Finally, we evaluated the performance of the selected model through training and validation.

Mathematical models have been developed to describe cancer cell metabolism from different perspectives. For example, Mendoza-Juez et al. [11] focused on glucose and lactate as the main nutrient resources, and thus concentrated on the dynamic development of two subpopulations with different metabolic behavior. Conversely, Astanin and Preziosi [12], while similarly modeling oxidative and glycolytic subpopulations, also included oxygen consumption and ATP production in their model. Without measurement, this system was characterized with typical values of dimensionless parameters for simulation. These models, with a heavy reliance on a large number of unmeasured parameters, can be difficult to calibrate and therefore difficult to apply within an experimental-predictive framework.

The family of models proposed in this study was derived from the work of Mendoza-Juez et al. [11], but included simplifications. We viewed the live tumor cells as a single population instead of two subpopulations with different metabolic phenotypes. In our model, we assumed the tumor cells may include all states varying between complete oxidative phosphorylation and complete anerobic glycolysis, instead of merely modeling the two phenotypes. This is most easily seen in Eq [1], where we treat the proliferation (growth) rate as the maximum rate possible under any metabolism. When the fraction of oxidative phosphorylation and anerobic glycolysis varies in response to different glucose levels, the modified real-time proliferation rate represents a weighted-average of the proliferation rates under complete oxidative phosphorylation and complete anerobic glycolysis. Therefore, we decided to focus on the overall cell population, rather than each cell's individual metabolic phenotype. In our model, the proliferation rate can be considered as an average of all cells possible proliferative activity (similarly for the death rate). Our simplification avoided the need to monitor the conversion between the two phenotypes as well as the fractional change over time. This approach has both experimental and computational advantages. Experimentally, we are not forced to develop a method that is capable of making serial, non-destructive measurements of metabolic activity of the same cells over time. Computationally, this approach reduces the number of parameters that we would need to calibrate from experimental data. At the cost of losing detailed

phenotype or subpopulation dynamics, the simplification allowed us to practically connect the accessible experimental data and mathematical modeling. Given sufficient experimental data (*via* the time-resolved microscopy data) and known initial conditions, we were able to perform a direct calibration of our model. This experimental-computational approach was applied in two cell lines, representing very different breast cancer subtypes. We found the proliferation rate of the BT-474 cells was statistically lower than MDA-MB-231 cells. We found the death rate due to glucose depletion for the BT-474 cells was statistically higher than that of the MDA-MB-231, while the consumption rate of glucose for the BT-474 cells is statistically lower than that of the MDA-MB-231 cells. We concluded that while MDA-MB-231 cells consume glucose at a higher rate (thereby enabling more rapid growth and division), the glucose level required for proliferation was lower than that of the BT-474 cells. These results serve to quantify the well-established experimental observations that MDA-MB-231 is more aggressive than the BT-474 cell line [43–45]. Once calibrated, our model could be used to predict the number of live tumor cells, validated by direct comparison with experimental data.

We noticed that the term describing the death due to glucose depletion was not sufficient to capture all the death observed in the experimental data. Considering the competition for space between the dead (prior to detaching) and live cells and, therefore, the potential cytotoxicity from the dead to the live cells, we introduced an extra term to account for this bystander effect. It represented the bulk phenomenon that dead cells can release factors, which may be sensed by the remaining live cells, and potentially induce cell death [26, 27]. We note that the bystander effect is multi-factorial as it involves multiple mechanisms, either mediated by GJIC (gap junction intracellular communication) capacity [46–48] or soluble factors [49–51]. The death rate due to the bystander effect, $k_{bys}$, proved to be dependent on the initial conditions. Therefore, $k_{bys}$ was estimated individually for each set of initial conditions, and not considered as a global parameter. The bystander effect parameter for both cell lines became significantly lower when the initial glucose level increased. This parameter significantly increased when the initial confluence for the BT-474 cell line increased, but was not affected by initial confluence in the MDA-MB-231 line. This difference indicates there could be different mechanisms underlying the bystander effect in different cell types. Studies concerning the bystander effect can be identified into two separate groups. In the first case, the bystander effect is proven to be mediated by degree of GJIC capacity [46–48]. Since BT-474 cells are mass cells with robust cell-cell adhesion and close cell contact within clusters, they have high GJIC capacity. However, MDA-MB-231 cells do not form clusters with strong cell contact and exhibit low GJIC level. These are consistent with the results that the death rate of bystander effect for BT-474 increases with initial confluence, but is not affected by initial confluence for MDA-MB-231. In the second case, killing of the non-treated cells involves the release of one or more soluble factors, such as apoptosis inducing signals [49], extracellular vesicles [50], or oxidizing diffusive factors [51]. In our study, the death rate of bystander effect for MDA-MB-231 is not affected by initial confluence, implying there would be at least one soluble factor regulated by metabolism. Furthermore, there could be multiple mechanisms underlying the bystander effect for a given cell line, considering the death rate of bystander effect for BT-474 is affected by both confluence and glucose level. While further work including the identification and quantification of these factors is required to support our work, this experimental-computational approach allows us to analyze the characteristics of bystander effect for the cell line tested. This could provide guidance on choice of enhanced therapies utilizing the bystander effect (e.g., GJIC enhancement) for synergistic effect [48].

The experimental-computational approach was developed and validated in two commonly studied breast cancer cell lines and is applicable to other cell lines. Given a new cell line, the same approach can be followed to estimate the proliferation rate, the death rate due to glucose

depletion, and the consumption rate of glucose, as well as determining the correlation between the death rate due to bystander effect and the initial conditions. Additionally, the framework allows for the prediction of the growth of a new tumor cell line given the initial conditions of glucose concentration and cell number. Thus, our experimental-mathematical approach allows for the systematic investigation of the response of different cell lines to glucose availability, thereby enabling the ability to quantitatively study the potential metabolism-related therapy.

The present work assumed glucose consumption was entirely captured by the temporal change of tumor cell number, which is most likely an oversimplification. To address this limitation, the development of a method for time-resolved measurement of glucose dynamics is required. Further quantification and mathematical description of the glucose dynamics (e.g., a FRET nanosensor for glucose [52]) would provide additional time-resolved data that would enable extension of the model to more precisely describe glucose kinetics. The hyperbolic tangent function introduced in Eqs [4] and [5] is sufficient to characterize the growth curves in our current research and we chose to keep it simple to avoid overfitting. However other sigmoidal functions of time that are more directly related to phenomena affecting glucose dynamics should be explored to refine to the model by introducing more biology. In particular, the state functions (i.e., Eqs [4] and [5]) have the potential to be extended to characterize glucose utilization as a function of cell cycle [53–55]. Additionally, our model could then be extended to account for additional nutrients of metabolic interest (e.g., lactate, intermediate products between glycolysis and oxidative phosphorylation, and oxygen). Such an extension would, of course, require additional time resolved measurements to parameterize the model.

Another area for investigation is in extending the present paradigm to 3D as there is (of course) a gap between well-controlled 2D monolayer and 3D cell cultures. In 3D, BT-474 cells form colonies with round borders, while MDA-MB-231 cells present an invasive phenotype with stellate projections that often bridge multiple cell colonies [25]. BT-474 cells develop dense multicellular spheroids (MCSs) in 3D cell culture, while MDA-MB-231 cells develop only loosely aggregated MCSs [56]. These different architectures can lead to different microenvironments for the tumor cells, resulting in a non-uniform distribution of nutrients like glucose and oxygen. For example, hypoxia areas can be observed inside the dense 3D-MCSs from the BT-474 cell lines, but not in the loosely aggregated MCSs from the MDA-MB-231 cells or cells in 2D-culture [56]. This suggests a higher fraction of glycolysis for BT-474 cells compared to MDA-MB-231 cells, which indicates that we need to include cell line dependent parameters for nutrient diffusion, as the diffusion rate can depend on cellular density and tumor architecture. In addition, as hypoxia has been reported to cause cancer cell dormancy in the G0 phase [57], a reduced proliferation rate can be expected for BT-474 cells. Thus, when applying our modeling approach to 3D data, with more complicated microenvironments and tumor architectures, we would likely need to design new experiments to characterize the spatial distribution of the cells and nutrients. We also limited the application of our model to only two different breast cancer cell lines, but given their differences in parameter values, systematic investigation of a range of cells lines is warranted.

In summary, the temporal change of tumor cell number with different initial glucose levels and seeding densities was tracked with time-resolved microscopy. These data were used to calibrate a mathematical model describing cell proliferation and death as a function of glucose dynamics, which was then used to predict tumor cell dynamics in a separate validation set. This approach yielded an accuracy of $> 75\%$ for predicting the change in the number of living cells over time, and is readily extendable to account for and predict the effects of interventions designed to affect glucose metabolism.

## 5. Conclusion

We have developed and validated an experimental-mathematical approach that is capable of accurately predicting how glucose availability influences tumor cell proliferation. The approach was validated in two commonly studied breast cancer cells in which we were able to quantify rates directly reporting on cell proliferation, death due to glucose starvation, death due to the bystander effect, and overall glucose consumption. The different relationships between $k_{bys}$ and the initial conditions found through model calibration suggested different mechanisms were involved in the bystander effect in these two breast cancer cell lines. The complete model, characterized by Eqs [1]–[5] featuring three global parameters ($k_p$, $k_d$, and $v$), and one local parameter dependent on initial conditions ($k_{bys}$), was able to provide the best characterization of the data, as indicated by the lowest AICc value. Furthermore, this frame-work is directly applicable to other tumor cell lines. The integration of mechanism-based modeling and time-resolved microscopy is a powerful, and flexible, approach to systematically investigate glucose dynamics related tumor cell growth. In addition, we could perform synthetic studies with our model to guide experimental design [58]. By evaluating the value of collecting glucose data at specific time points, we will be able to further validate and optimize the current model with extra data inputs for future work.

## Supporting information

**S1 Fig. Steps on cell segmentation for phase-contrast images of BT-474 breast cancer cell lines.** The size of the whole well image is 2400 x 2400 pixels. Here we present a window of 400 x 400 pixels from an example image. Panel A: raw image of BT-474 cells; Panel B: image post 'colfilt'; Panel C: image post the Gaussian filter; Panel D: image post 'im2bw'; Panel E: image post 'imerode'; Panel F: image post 'imclose'; Panel G: image post 'imopen'; Panel H: image post 'bwareaopen', the final cell mask; Panel I: overlay of raw image and the cell mask for BT-474 cells.
(TIF)

**S2 Fig. Steps on cell segmentation for phase-contrast images of MDA-MB-231 breast cancer cell lines.** The size of the whole well image is 2400 x 2400 pixels. Here we present a window of 400 x 400 pixels from an example image. Panel A: raw image of MDA-MB-231 cells; Panel B: image post binarization; Panel C: image post 'imclose'; Panel D: image post 'bwareaopen', the final cell mask; Panel E: overlay of raw images and the cell mask for MDA-MB-231 cells.
(TIF)

**S3 Fig. Time courses of tumor cell confluence in media with 0 mM glucose, grouped by initial confluence.** Tumor cells may keep proliferating for some time even in a glucose free medium, even up to 24 hours for MDA-MB-231 (Panel A). The proliferation in glucose free medium is not observed for BT-474 (Panel B).
(TIF)

**S4 Fig. Estimates of the death rate due to the bystander effect, $k_{bys}$, as a function of different initial confluence, for BT-474 cells.** Each subtitle indicates the initial glucose concentration. For a given initial glucose level, $k_{bys}$ increases with initial confluence. For 8 of 10 initial glucose levels tested (0, 0.1, 0.2, 0.5, 0.8, 1, 2, and 5 mM, the bystander effect death rate is positively correlated with initial confluence, with correlation coefficients all $> 0.74$ ($p < 0.01$). For the highest two initial glucose levels (8 and 10 mM), there is no significant correlation between the bystander effect death rate and the initial confluence ($p > 0.1$).
(TIF)

**S5 Fig. Estimates of the death rate due to the bystander effect, $k_{bys}$, as a function of different initial confluence, for MDA-MB-231 cells.** Each subtitle indicates the initial glucose concentration. There is no significant correlation between the bystander effect death rate and the initial confluence (p > 0.1).
(TIF)

**S6 Fig. Model predictions for BT-474 cells.** Example model predictions of glucose levels from one validation set of BT-474 cells. The average glucose levels from predictions are shown as dashed lines, with the 95% confidence intervals shown as shaded regions between the solid curves. The initial glucose level is shown above each plot. Please note the scales of vertical axis in each panel are different to better visualize the change of glucose levels.
(TIF)

**S7 Fig. Model predictions for MDA-MB-231 cells.** Example model predictions of glucose levels from one validation set of MDA-MB-231 cells. The average glucose levels from predictions are shown as dashed lines, with the 95% confidence intervals shown as shaded regions between the solid curves. The initial glucose level is shown above each plot. Please note the scales of vertical axis in each panel are different to better visualize the change of glucose levels.
(TIF)

**S1 Table. Results of AICc value for model selection.** Model 2 is the complete model described by Eq [1]–[5]. In Model 1, all the terms involving $k_{bys}$ is removed, while in Model 3, any term involving $k_d$ is removed. In the first calibration, the measured live and dead tumor cell time courses are independently fit to the model to produce separate estimates for each model parameter. In the second calibration, all the parameters are treated as global parameters. In the third calibration, $k_{bys}$ is considered as a local parameter while the other parameters ($k_p$, $k_d$, and $v$) are treated as global parameters.
(DOCX)

**S2 Table. Explicit matrices of variables used to evaluate the model's performance.** $X_{model,ij}$ is the number of live or dead cells of well $j$ at timepoint $i$ calculated from the model, $X_{data,ij}$ is the number of live or dead cells of well $j$ at timepoint $i$ from the measured data, $t$ is the total number of timepoints, $w$ is the total number of wells, and $t_{end}$ is the last timepoint at the end of experiment (EoE).
(DOCX)

**S3 Table. Relative contributions of each term in the mathematical model for the BT-474 cells.** The table shows the relative contributions of each term of Eq [1] on days 0, 2, and 4 for different initial confluences and three initial glucose concentrations (0, 1, and 10 mM).
(DOCX)

**S4 Table. Relative contributions of each term in the mathematical model for the MDA-MB-231 cells.** The table shows the relative contributions of each term of Eq [1] on days 0, 2, and 4 for different initial confluences and three initial glucose concentrations (0, 1, and 10 mM).
(DOCX)

## Acknowledgments

We are grateful for the members of the Center for Computational Oncology at The University of Texas at Austin for many helpful and informative discussions. We thank Dr. Heiko Enderling and Dr. Chad Quarles for constructive feedback at the early stages of this project.

## Author Contributions

**Conceptualization:** Jianchen Yang, Jack Virostko, Thomas E. Yankeelov.

**Data curation:** Jianchen Yang.

**Formal analysis:** Jianchen Yang.

**Funding acquisition:** Thomas E. Yankeelov.

**Investigation:** Jianchen Yang.

**Methodology:** Jianchen Yang, David A. Hormuth, II, Thomas E. Yankeelov.

**Project administration:** Thomas E. Yankeelov.

**Software:** Jianchen Yang, Junyan Liu.

**Supervision:** Jack Virostko, Amy Brock, Jeanne Kowalski, Thomas E. Yankeelov.

**Validation:** Jianchen Yang.

**Writing – original draft:** Jianchen Yang, Thomas E. Yankeelov.

**Writing – review & editing:** Jianchen Yang, Jack Virostko, David A. Hormuth, II, Amy Brock, Thomas E. Yankeelov.

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
