## [Decision Letter · Decision Letter 0]

19 Nov 2020

PONE-D-20-30580

An experimental-mathematical approach to predict tumor cell growth as a function of glucose availability in breast cancer cell lines

PLOS ONE

Dear Dr. Yankeelov,

Thank you for submitting your manuscript to PLOS ONE. After careful consideration, we feel that it has merit but does not fully meet PLOS ONE’s publication criteria as it currently stands. Therefore, we invite you to submit a revised version of the manuscript that addresses the points raised during the review process.

In particular, the proposed experimental-computational approach is very interesting but the modelling assumptions need to be justified and detailed more. We would also like to encourage you to include more details on data availability.  

We look forward to receiving your revised manuscript.

Kind regards,

Aurélie Carlier

Academic Editor

PLOS ONE

2. Can the code used in the study be shared with other researchers? If so, please indicate where the code can be found in your Data availability statement.

Reviewers' comments:

Reviewer's Responses to Questions

**Comments to the Author**

1. Is the manuscript technically sound, and do the data support the conclusions?

Reviewer #1: Partly

Reviewer #2: Partly

2. Has the statistical analysis been performed appropriately and rigorously? 

Reviewer #1: Yes

Reviewer #2: N/A

3. Have the authors made all data underlying the findings in their manuscript fully available?

Reviewer #1: Yes

Reviewer #2: No

4. Is the manuscript presented in an intelligible fashion and written in standard English?

Reviewer #1: Yes

Reviewer #2: Yes

5. Review Comments to the Author

Reviewer #1: This is an interesting paper focusing on fitting different models to the experimental data on the temporal change of the number of live and dead tumor cells under different initial glucose concentrations and seeding densities. However, I have number of concerns regarding the modelling, which I believe have to be addressed.

Major concerns:

1. Unreadable mathematical expressions: Practically all equations ([1]-[6]) in the manuscript are unreadable , as many squares and rectangles appear instead of signs +,-, and =, and possibly also brackets. As a consequence, the rationale behind those equations cannot be checked, which makes it impossible for me to give my full judgement of the models used. This needs to be corrected.

2. Unmotivated manipulation of death rates as opposed to growth rates: I do not understand why when you experimentally manipulate the glucose concentration which should contribute to faster proliferation, your models do not differ in their growth rate, but instead in their death rate. Please explain or switch to more standard interpretation where glucose contributes to faster proliferation of cells.

3. Unmotivated simplification of the model by Mendoza-Juez et al: It is unclear why the authors do not use 2 phenotypes as Mendoza-Juez et al. but instead assume that all cells use anaerobic glycolysis.

Is that inherent to the cell lines used? Is it 100% sure that no cells use oxidative phosphorylation?

If not, why isn't it better to estimate the subpopulations pf Mendoze-Juez as opposed to removing one type? (Here I note that my problem with this may be a consequence of inability to read the equations, as mentioned in 1., thus correction of that problem as asked in 1. may help me to understand the simplification better)

4. Strong dependence on the results on the percentage of training and validation data: The authors report that with the selected model, the achieved accuracy (defined as the fraction of

37 measured data that fell within the 95% confidence intervals of the predicted growth curves) is 77.2

38 ± 6.3% and 87.2 ± 5.1% for live BT-474 and MDA-MB-231 cells, respectively. This accuracy will surely depend on the percentage of data used for training/validation. Assuming that there are reasons for having the model form the authors use (see concerns I express in my previous points), could the authors explain why exactly 75% of data are used for training (and not more or less)? And if not, could they perform sensitivity analysis with respect to this percentage? Also, I do not perceive 77.2 % as particularly high. Could the authors comment why they think their model outperforms other, perhaps more standard models when fitting the data?

Minor concerns:

5. I would be more modest in statements like this: "To the best of our knowledge, this represents the first time these important cellular parameters have been quantified within a rigorous modeling framework." I know of many studies where cellular parameters are quantified with a modeling framework (eg Kaznatcheev et al, Nature Eco Evo 3(3), 2019; or works by Brady and Enderling....), and I am sure the authors know many, too.

6. It is interesting to see a paper on Warburg effect which does not mention a large body of literature of game theoretical models that attempt to explain/quantify it, for example Archetti at al, Epstein et al. I would recommend adding these references.

7. "The major source of energy for most cancer cells comes from a high rate of glycolysis

followed by lactate fermentation in the cytosol, even in the presence of sufficient oxygen—a

phenomenon known as the Warburg effect [1,2]." - Are you sure of the word "most" here?

Reviewer #2: It is not clear to me that the authors have made their data available

I would encourage them to include details explaining how people can access the data

and use it to perform their own analysis of the data.

---

## [Author Response · Author response to Decision Letter 0]

5 May 2021

We thank the reviewers for their careful review of our manuscript and for providing such constructive criticism. We have taken their suggestions and criticisms seriously, and have revised the manuscript to address their concerns. As a result of their feedback, we believe the quality and clarity of our manuscript has substantially improved. Detailed responses to each item are provided in the "Response to Reviewers" documents.

---

## [Decision Letter · Decision Letter 1]

18 Jun 2021

PONE-D-20-30580R1

An experimental-mathematical approach to predict tumor cell growth as a function of glucose availability in breast cancer cell lines

PLOS ONE

Dear Dr. Yankeelov,

Thank you for submitting your manuscript to PLOS ONE. After careful consideration, we feel that the reviewer comments were addressed and that the manuscript can be accepted pending the small suggested revisions below.

We look forward to receiving your revised manuscript.

Kind regards,

Aurélie Carlier

Academic Editor

PLOS ONE

Journal Requirements:

Reviewers' comments:

Reviewer's Responses to Questions

**Comments to the Author**

1. If the authors have adequately addressed your comments raised in a previous round of review and you feel that this manuscript is now acceptable for publication, you may indicate that here to bypass the “Comments to the Author” section, enter your conflict of interest statement in the “Confidential to Editor” section, and submit your "Accept" recommendation.

Reviewer #2: All comments have been addressed

Reviewer #3: (No Response)

2. Is the manuscript technically sound, and do the data support the conclusions?

Reviewer #2: Yes

Reviewer #3: Yes

3. Has the statistical analysis been performed appropriately and rigorously? 

Reviewer #2: Yes

Reviewer #3: Yes

4. Have the authors made all data underlying the findings in their manuscript fully available?

Reviewer #2: Yes

Reviewer #3: Yes

5. Is the manuscript presented in an intelligible fashion and written in standard English?

Reviewer #2: Yes

Reviewer #3: Yes

6. Review Comments to the Author

Reviewer #2: The authors have done a good job of revising their manuscript in line with my comments and those of the other reviewer. As a result, I am happy to recommend it for publication.

I have only one minor request for further clarification: on page 15, the authors talk about 'reliance of the model parameters on the initial conditions' - what does this mean? that the model parameters are correlated with the initial conditions?

Reviewer #3: Overview:

The authors present a combined experimental and modeling approach to determine rates of proliferation, death, and glucose consumption for tumor cells, in varying glucose conditions. The paper achieves this goal, and the results are clearly presented. Though I did not review the first version, I have read the reviewers' comments and the authors' responses. I believe the authors have addressed the previous concerns.

Specific comments:

Minor points:

(1) the equations (still?) do not appear correctly, with operation signs (addition, multiplication) missing. I assume this will be corrected in the proofing stage.

(2) in the text, p values should be shown as numerical values (10^-4) rather than "1e-4".

Larger points:

(1) My primary major concern is in the presentation of the utility of the work. I would like to see some explanation about the impact and utility of the modeling work. In its current state, the paper does a great job in presenting the model and results. However, how others could take advantage of the work and use it to gain biological insight is missing. I feel this should be addressed in order to increase the impact of the work. How would the model do with a different cell line, say MCF7?

(2) The abstract, intro, and methods emphasize the family of models and three approaches for calibration, but all of that (interesting stuff!) is relegated to a single table in the supplemental info (S1 Table). I would prefer to see a bit more description of those steps in the main text, as the work raises some interesting questions. For example, why do the model that neglects the bystander effect or the model that neglects death due to low glucose NOT fit the data well? These mechanistic questions are of interest, especially when considering the previous point - the utility of the work. If this is not the focus, but rather the goal is to show results from a single model, the intro and methods should be revised.

7. PLOS authors have the option to publish the peer review history of their article (what does this mean?). If published, this will include your full peer review and any attached files.

Reviewer #2: No

Reviewer #3: No

---

## [Author Response · Author response to Decision Letter 1]

26 Jun 2021

Manuscript #PONE-D-20-30580R1

We thank the reviewers for their careful consideration of our revised manuscript and for providing more constructive criticism. We have taken these comments seriously and have revised the manuscript accordingly. Detailed responses to each item are provided below. In our responses below, for clarity, we have referred to Review x, Comment y as ‘Rx.y’.

Editor’s Comments:

Thank you for submitting your manuscript to PLOS ONE. After careful consideration, we feel that the reviewer comments were addressed and that the manuscript can be accepted pending the small suggested revisions below.

Response:

We sincerely thank the Editor for their comments on the revision. As indicated below (and in the revised manuscript), we have worked hard to address each suggested revision.

Specific review questions

1. If the authors have adequately addressed your comments raised in a previous round of review and you feel that this manuscript is now acceptable for publication, you may indicate that here to bypass the “Comments to the Author” section, enter your conflict of interest statement in the “Confidential to Editor” section, and submit your "Accept" recommendation.

Reviewer # 2: All comments have been addressed.

Response:

We thank the reviewer for their support.

Reviewer #3: (No Response)

Response:

N/A

Reviewer 2 individual comments

R2.0) The authors have done a good job of revising their manuscript in line with my comments and those of the other reviewer. As a result, I am happy to recommend it for publication.

Response: 

We sincerely thank the reviewer for their support.

R2.1) I have only one minor request for further clarification: on page 15, the authors talk about 'reliance of the model parameters on the initial conditions' - what does this mean? that the model parameters are correlated with the initial conditions?

Response: 

We apologize for the lack of clarity on this point. Yes, we intended to investigate the correlation between the model parameters and the initial conditions. (The justification for model parameters depending on initial conditions can be found in response to R2.8 in the initial ‘response to reviewers’ document and the revised manuscript.) We have rephrased the text on page 15 to provide more clarification on this point. 

Reviewer 3 individual comments

R3.0) The authors present a combined experimental and modeling approach to determine rates of proliferation, death, and glucose consumption for tumor cells, in varying glucose conditions. The paper achieves this goal, and the results are clearly presented. Though I did not review the first version, I have read the reviewers' comments and the authors' responses. I believe the authors have addressed the previous concerns.

Response: 

We sincerely thank the reviewer for their support.

R3.1) The equations do not appear correctly, with operation signs (addition, multiplication) missing. I assume this will be corrected in the proofing stage.

Response: 

We apologize for this unfortunate situation and will work with the editor for correction in the proofing stage.

R3.2) In the text, p values should be shown as numerical values (10^-4) rather than “1e-4”.

Response: 

We have changed the way we present the p values the revised manuscript to align with the reviewer’s suggestion throughout.

R3.3) My primary major concern is in the presentation of the utility of the work. I would like to see some explanation about the impact and utility of the modeling work. In its current state, the paper does a great job in presenting the model and results. However, how others could take advantage of the work and use it to gain biological insight is missing. I feel this should be addressed in order to increase the impact of the work. How would the model do with a different cell line, say MCF7?

Response: 

We apologize for the lack of explanation for the impact and utility of the modeling work. The experimental-computational approach was developed and validated in two commonly studied breast cancer cell lines and is applicable to other cell lines. Given a new cell line, say MCF7, the same approach can be followed to estimate the proliferation rate, the death rate due to glucose depletion, and the consumption rate of glucose, as well as determining the correlation between the death rate due to the bystander effect and the initial conditions. Additionally, the framework allows for the prediction of the growth of a new tumor cell line given the initial conditions of glucose concentration and cell number. Thus, our experimental-mathematical approach allows for the systematic investigation of the response of different cell lines to glucose availability, thereby enabling the ability to quantitatively study the potential metabolism-related therapy. We have revised the Discussion section to address the impact and utility of the modeling work. 

R3.4) The abstract, intro, and methods emphasize the family of models and three approaches for calibration, but all of that (interesting stuff!) is relegated to a single table in the supplemental info (S1 Table). I would prefer to see a bit more description of those steps in the main text, as the work raises some interesting questions. For example, why do the model that neglects the bystander effect or the model that neglects death due to low glucose NOT fit the data well? These mechanistic questions are of interest, especially when considering the previous point - the utility of the work. If this is not the focus, but rather the goal is to show results from a single model, the intro and methods should be revised.

Response: 

We apologize for the lack of description of the other two members of the family of models and have added more details in the main text of the revised manuscript. The other two models did not perform as well as the model that included both death terms, as they yielded a significantly higher fitting error to the data. For model 1, the model that neglects the bystander effect, the death due to glucose depletion is not sufficient to capture all of the observed cell death, especially for cells seeded with high initial glucose and high initial confluence. For model 3, the model that neglects the death due to glucose depletion, the initiation of cell death cannot be explained when there are initially no dead cells and, therefore, perform worse and overestimate the death rate for cells with low glucose level and few dead cells at the beginning. We have added these comments to the revised Discussion section.

---

## [Editor Report · Decision Letter 2]

28 Jun 2021

An experimental-mathematical approach to predict tumor cell growth as a function of glucose availability in breast cancer cell lines

PONE-D-20-30580R2

Dear Dr. Yankeelov,

We’re pleased to inform you that your manuscript has been judged scientifically suitable for publication and will be formally accepted for publication once it meets all outstanding technical requirements.

Kind regards,

Aurélie Carlier

Academic Editor

PLOS ONE
---

## [Editor Report · Acceptance letter]

5 Jul 2021

PONE-D-20-30580R2 

An experimental-mathematical approach to predict tumor cell growth as a function of glucose availability in breast cancer cell lines 

Dear Dr. Yankeelov:

I'm pleased to inform you that your manuscript has been deemed suitable for publication in PLOS ONE. Congratulations! Your manuscript is now with our production department. 

Kind regards, 

on behalf of

Dr. Aurélie Carlier 

Academic Editor

PLOS ONE